# Perceived Importance of Smart and Sustainable Building Features from the Users' Perspective

**Wai Ming To [1],\*** , **Linda S. L. Lai [1]**, **King Hang Lam [2]** and **Andy W. L. Chung [3]**

1   School of Business, Macao Polytechnic Institute, Macao, China; sllai@ipm.edu.mo
2   Department of Electrical and Electronic Engineering, The University of Hong Kong, Hong Kong, China; khlam@eee.hku.hk
3   Macau Instituto de Acustica, Macao, China; andy@moiacoustics.org
\*   Correspondence: wmto@ipm.edu.mo; Tel.: +853-8599-3319

**Abstract:** Smart and sustainable buildings have been designed, built and utilized in order to consume less energy, facilitate efficient building operation, and improve the comfort, health and productivity of users. Hence, they become a critical component of smart cities. Nonetheless, perceived importance rankings of different features of smart and sustainable buildings have yet to be identified and prioritized from the users' perspective. Based on responses from 494 building users in Hong Kong, it was found that building users tended to focus more on intelligent security systems, followed by intelligent and responsive fresh air supply and lifts and escalators. On the other hand, building users generally considered the systems that monitor people's movement and harvest rain water to be the least important features. Exploratory factor analyses were used to identify key factors of perceived smartness and sustainability of a building. The results of factor analyses showed that different user groups would characterize a building's smartness differently.

**Keywords:** smart and sustainable buildings; importance ratings; exploratory factor analysis; users' perspective

## 1. Introduction

Smart and sustainable buildings have attracted considerable attention in recent years. According to a report published by BSRIA [1], the smart building market alone was projected to grow from US\$427 billion in 2011 to US\$1036 billion by 2020, thereby creating vast opportunities to develop and apply advanced building technologies, and information and communications technologies (ICT). China will account for a 46% market share in the global smart building industry [1]. Smart and sustainable buildings are the ones that through their physical design and ICT installations are responsive and adaptive to the changing environment and needs of users throughout their lifetimes. Because of the adaptive nature of smart and sustainable buildings, the usage of materials, water, and energy resources can be optimized while comfortable and healthy indoor environments can be achieved [2–4]. As a building block of smart cities, smart and sustainable buildings promote sustainable urbanization, healthy life style, and social inclusion [5–8].

Although the concept of smart and sustainable buildings is not new, the development of smart and sustainable buildings is now becoming a reality because of the emergence of smart sensors, smart appliances, smart systems, cloud computing, and ubiquitous connectivity. In fact, building designers, engineers, and researchers have designed and promoted smart and sustainable buildings and their predecessors, intelligent buildings, for decades [2,9,10]. In the early 1990s, studies on intelligent buildings focused mainly on developing advanced control systems for heating, ventilation and air conditioning, lighting, lifts and escalators, fire-fighting, security, telecommunication

and data services, and on integrating these control systems into a holistic building management system. Building designers, engineers, and researchers advocated for all these ideas, and, consequently, the interactivity of a building with its users and between users were kept minimal. In that regard, intelligence means making decisions on behalf of building users. The concept of intelligent buildings gradually expanded over time [9,11–13]. Clements-Croome [14] suggests that intelligent buildings must be responsive to the requirements of occupants, organizations, and society. Such buildings must also be sustainable because of the minimized use of energy and water resources as well as the reduced emissions of pollutants and waste. Thus, smart and sustainable buildings must promote the well-being of users, must adapt to their functional needs, and keep the impact on the environment minimal [3,14–16]. Yet, the extant literature on smart and sustainable buildings predominantly focuses on technological details; there is scant literature to adopt the user-centric approach to evaluate the smartness and sustainability of buildings. Hence, the study aims to address the following three research questions: (i) How do building users evaluate the importance of different features of smart and sustainable buildings? (ii) What are the key factors of perceived smartness and sustainability from the users' perspective? (iii) Are the perceptions of building smartness and sustainability different among different user groups? The study's results can provide important insights that allow building designers and owners to focus on what users really expect from smart and sustainable buildings.

## 2. Literature Review

A smart and sustainable building should definitely be intelligent. Nevertheless, a building's excessive intelligence does not only minimize human interaction with the building's systems but also dictates how the systems should be operated, possibly causing frustration among users [2,17]. Thus, a smart and sustainable building must provide users with the "appropriate" services (in view of its intelligence) and information [2,16]. The building's environment shall adapt to most users' needs but it also allows individual users to adjust the settings according to his/her individual needs and preferences at ease i.e., responsive to user demands [15,16].

Buckman et al. [2] defined a smart and sustainable building as a structure that integrates intelligence from sensing elements and user demands, different control strategies, materials and construction as an integrated building system, with adaptability, not reactivity, at its core. The three basic drivers for designing a smart and sustainable building are energy and efficiency, longevity and evolving capabilities to sustain the building over its life cycle, and users' comfort and satisfaction [2,13]. Arditi et al. [16] explored the degree of smartness of a building. Based on responses from 120 building designers, contractors and owners, Arditi et al. [16] developed a smartness index that covers economic issues, energy issues, and comfort issues. Arditi et al. [16] found that building designers, contractors, and owners emphasized more on energy issues over the other two issues. Moreover, numerous people now spend more than 8 hours a day for 5 days a week (i.e., a total of 48 weeks in a year) in commercial and office buildings. Thus, the social functions of smart and sustainable buildings should be taken into consideration [18]. In addition, the greening of buildings (or the city) can promote harmony to urban life and regulate the microclimate [19,20].

To et al. [4] studied the perceived importance of a wide range of smart and sustainable building features. They reported that an intelligent security system, fresh air supply, and thermal control were rated as the top three most important features from building professionals' and designers' perspective. Surprisingly, indoor green and social spaces received relatively low importance ratings from building professionals and designers. To et al. [4] also applied the extended Technology Acceptance Model to identify what drives building professionals and designers to adopt smart and sustainable building technologies. They reported that perceived ease of use, perceived usefulness, and facilitating conditions induced building professionals and designers to have a high level of intention to use smart and sustainable building technologies.

*Importance Ratings of Key Features*

The use of importance ratings to identify users' perceptions is well established [4,21–24]. Marshall et al. [21] indicated that a firm's internal service quality affects employee satisfaction, which in turn influences external service quality, customer satisfaction, and the profitability of the firm. Based on interviews with 83 employees in a manufacturing firm, Marshall et al. [21] developed 24 items covering a wide range of service features that were viewed as generally important to internal users. After obtaining 114 usable questionnaires from another group of users, Marshall et al. [21] applied factor analysis to importance ratings and obtained a six-factor structure of internal service quality in the manufacturing firm. Iseki and Taylor [22] explored user perceptions of mass transit stations in Los Angeles. They focused on the relative importance that users place on different aspects of their wait/walk/transfer experience at mass transit stations, and user satisfaction with each of these aspects. The importance ratings showed that security and safety were the most important features in determining users' station experience. Lai and Yik [23] obtained importance ratings from 297 users of a Hong Kong's residential estate to evaluate its facility management services. Lai and Yik [23] found that security had the highest mean importance ratings, followed by cleaning.

To et al. [24] explored the perceived importance and performance of environmental practices among Hong Kong's professional employees. They employed exploratory factor analysis to identify the underlying factor structure of environmental practices based on the importance ratings. Pottage and Jeffery [25] explored key features of sustainable buildings. Based on the importance ratings collected from office building users in the United Kingdom, Pottage and Jeffery [25] reported that daylight, indoor air quality, and indoor temperature were identified as influencing factors of office employees' well-being and productivity. To et al. [4] identified a wide range of smart and sustainable building features from the extant literature and investigated the relative perceived importance of the features from building professionals' and designers' perspective. In the present study, we adopted the items established by To et al. [4] and asked building users to evaluate the degree of importance from their perspectives.

## 3. Methods

In order to answer the three research questions, we adopted a cross-sectional survey study. We developed a self-administered questionnaire based on an extensive literature review. We used a chain referral sampling approach in which building users working in different industries were invited to participate in the survey voluntarily. Chain referral sampling has been widely used in exploratory studies across a wide range of contexts [26,27].

### 3.1. The Context of the Study

Hong Kong is an ideal city for building design studies because a large number of office buildings have been marketed as intelligent and smart over the past two decades [28]. Additionally, the Hong Kong Government is taking a proactive approach in developing Hong Kong into a smart city [29]. The Hong Kong Industry Associations have organized numerous conferences and exhibitions to promote the design and use of smart and sustainable buildings as well as smart and sustainable city over the past few years. In addition, the Hong Kong Smart City Consortium (https://smartcity.org.hk) was formed with the blessing from the Chief Executive of Hong Kong. Because of this recent development, many of Hong Kong's building users are aware of the characteristics of 'smart and sustainable buildings/city' such as their intelligence, adaptability, responsiveness, and environmental friendliness.

### 3.2. Design of the Questionnaire

The questionnaire has two parts. The first part consisted of 17 items that describe key features of a smart and sustainable building [2,4,9,14,16,18]. The items cover intelligent and responsive building elements such as lighting, fresh air supply, thermal control, acoustic element, expandable network infrastructure, security system, etc. [2,4,9,14,16]; responsive designs to cope with the changing external environment such as daylight, rain, smart grid, etc. [4,9,14,16]; and indoor ecology and social features [4,16,18]. All items were rated using a five-point Likert scale with 1 being "very unimportant" and 5 being "very important". More specifically, Table 1 shows the first part of the questionnaire including an introduction to the study, the core dimensions, the items used, and the rating scale. The second part consisted of six questions to collect respondents' demographic information like gender, age, education, industry, job position, and work experience.

**Table 1.** The items used in the study.

| Core Dimension | Item | Ref. |
|---|---|---|
| Building indoor environment | An intelligent and responsive fresh air supply is . . . <br> An intelligent and responsive thermal control is . . . <br> An intelligent and responsive lighting is . . . <br> An intelligent transport system such as lifts and escalators is . . . <br> A responsive acoustic environment is . . . | [2,4,9,14,16] |
| Smart building skin i.e., responsive to ambient environment | A responsive system that can harvest solar and wind energy is . . . <br> A responsive facade that can harvest daylight is . . . <br> A responsive system that can harvest rainwater is . . . <br> An intelligent system that can respond to smart grid is . . . | [4,9,14,16] |
| Eco and social spaces i.e., communal factor | An indoor social space is . . . <br> A real indoor green space with a variety of plants is . . . <br> Different social venues that facilitate interaction between users are . . . <br> Having minimum impacts on the environment is . . . | [4,16,18] |
| Building security and network systems | An intelligent security system is . . . <br> An intelligent system that monitors people movement is . . . <br> An expandable network infrastructure is . . . <br> A building information system is . . . | [2,4,9,14,16] |

Notes: The following introduction to the study and rating scale were used. Introduction: This questionnaire is designed to collect information about how you rate the importance of smart and sustainable building features. Please select the most appropriate answer for each question based on your own evaluation. The survey is anonymous and the data are kept confidential. Additionally, you can withdraw from the survey at any time while completing the questionnaire; A five-point Likert scale was used with 1 being "very unimportant" and 5 being "very important".

The questionnaire was pilot-tested on 10 building users who have knowledge about smart and sustainable buildings. The participants in the pilot study stated that items were clear and they completed the questionnaire in 10 to 15 min.

### 3.3. Data Collection Procedure

Although we only solicited respondents' importance ratings of different features of smart and sustainable buildings, we followed the standard research protocol approved by the University's ethical committee strictly. We explained clearly the purpose of the study in the questionnaire. As respondents were invited to participate in the survey voluntarily, they could withdraw from the survey at any time. All data were anonymous and kept confidential. We invited part-time Master's students from a university in Hong Kong who have full-time jobs to participate in the survey and requested them to distribute a total of 1000 paper-based questionnaires to their colleagues, friends, and relatives after the pilot test. After a 2-month period, we received 494 valid responses, representing a 49.4 percent usable response rate. All respondents were employees working in office environments.

*3.4. Data Analysis*

We used demographics to characterize the profile of respondents. The relative importance rankings of smart and sustainable building features were determined and categorized by gender and industry. A number of *t*-tests were performed to examine whether gender had any influence on the mean scores. Analyses of variance (ANOVAs) were performed to investigate industry as a significant between-group factor. We used exploratory factor analyses (EFA) to identify the factor structure of perceived smartness of smart and sustainable buildings from the users' perspective [30].

## 4. Results

Approximately 55 percent of the 494 respondents were males. The majority of the respondents (63.4%) were in the age group of 20–29. Most of the respondents (54.3%) had a bachelor's degree. Approximately 32 percent of the respondents worked in the engineering and construction services industry, followed by 20 percent working in the information and communications industry. All respondents had full-time jobs. About half of the respondents worked in a professional or senior position. Sixty-one percent of the respondents had 2 or more years of work experience. Table 2 shows the demographic profile of the respondents.

**Table 2.** Demographic characteristics of respondents (*n* = 494).

| Variable | Class | *n* | % | Variable | Class | *n* | % |
|---|---|---|---|---|---|---|---|
| Gender | Male | 272 | 55.1 | | Commerce & trad. | 65 | 13.2 |
| | Female | 222 | 44.9 | | Banking & finan. | 56 | 11.3 |
| Age | <20 | 14 | 2.8 | Industry | Inform. & comm. | 98 | 19.8 |
| | 20–29 | 313 | 63.4 | | Engg. & construct. | 157 | 31.8 |
| | 30–39 | 108 | 21.8 | | Others | 118 | 23.9 |
| | 40–49 | 32 | 6.5 | | Junior/frontline | 248 | 50.2 |
| | 50 or above | 27 | 5.5 | | Professional | 121 | 24.5 |
| Education | High school | 68 | 13.8 | Position | Senior staff | 57 | 11.5 |
| | Bachelor | 268 | 54.3 | | Managerial staff | 40 | 8.1 |
| | Masters | 133 | 26.9 | | Director | 28 | 5.7 |
| | Doctorate | 8 | 1.6 | | <1 year | 132 | 26.7 |
| | Others | 17 | 3.4 | | 1 to <2 years | 61 | 12.3 |
| | | | | Work Exp. | 2 to <4 years | 104 | 21.1 |
| | | | | | 4 to <8 years | 91 | 18.4 |
| | | | | | 8 years or more | 106 | 21.5 |

*4.1. Importance Features of Smart and Sustainable Buildings*

The means and standard deviations of all items were determined as shown in Table 3. Respondents indicated that "an intelligent security system" was the most important feature (mean = 4.16; SD = 0.99). The second and third most important features were "an intelligent responsive fresh air supply" and "an intelligent transport system (i.e., lifts and escalators)", respectively. On the other hand, respondents indicated that "an intelligent system that monitors people movement" was the least important feature (mean = 3.37; SD = 1.07). The second and third least important features were "a responsive system that can harvest rainwater" and "an intelligent energy system that can respond to smart grid", respectively. When the responsive acoustic environment is concerned i.e., using passive sound absorbers to control reverberation and active speakers for broadcasting sounds and background music, it was ranked 12th among the 17 items. Respondents also ranked indoor green space as the 8th important features, indoor social space and other social venues as the 10th and 13th important features. Thus, respondents focused more on security, good indoor air and their mobility in smart and sustainable buildings. Acoustic, green indoor and social environment could be considered as secondary features in smart and sustainable buildings from the users' perspective.

One-sample *t*-tests were performed to test whether the mean scores of the 17 items were higher than the mid-point of the scale (i.e., 3.0 in a five-point Likert scale). The results of all *t*-tests showed that all mean scores were significantly higher than 3.0 (all *p* values < 0.001). Independent-sample *t*-tests were performed to examine whether gender influenced the mean scores. The results showed that female respondents rated the level of importance higher than male respondents on 12 out of the 17 items. Among the 17 items, there were no significant differences between the mean ratings from male and female respondents on 10 items. Nevertheless, there were significant differences between the mean ratings from male and female respondents on seven items as shown in Table 3.

Table 4 shows the ranking of smart and sustainable building features categorized by the industry. Employees in different industries generally agreed that an intelligent security system, fresh air supply, and lifts and escalators were the three most important smart and sustainable building features while an intelligent system that monitors people movement and a responsive system that harvests rainwater were the two least important smart and sustainable building features. The item "a responsive acoustic environment is . . . " was ranked between 8 and 14 by respondents working in different industries. The item "an indoor social space is . . . " was ranked between 7 and 13 while the item "different social venues that facilitate interaction between users are . . . " was ranked between 10 and 14 by respondents working in different industries. The item "a real indoor green space with a variety of plants is . . . " was ranked between 4 and 15 by respondents working in different industries. It was found that employees working in the information and communications industry paid a greater attention to the building information system in their importance ranking in comparison with that of other groups. The ANOVA results indicated that there were significant differences in mean values among 14 out of 17 smart and sustainable building features between respondents working in different industries, except three features covering "an expandable network infrastructure", "a building information system", and "an intelligent system that monitors people movement". The mean importance ratings of respondents working in the information and communications industry were significantly lower than that of the engineering and construction services industry and the "others". A more detailed observation of the ANOVA results showed that there were no significant differences between responses from employees working in the commerce and trading industry and in the banking and finance industry and between those working in the engineering and construction services industry and the "others".

**Table 3.** The overall mean importance ratings of smart and sustainable buildings' features (from high to low).

| Items | Overall *n* = 494 | | | Male *n* = 272 | | | Female *n* = 222 | | | Gender Difference |
|---|---|---|---|---|---|---|---|---|---|---|
| | Mean | S.D. | Rank | Mean | S.D. | Rank | Mean | SD | Rank | (*p*-Value) |
| An intelligent security system is . . . | 4.16 | 0.99 | 1 | 4.13 | 1.00 | 1 | 4.20 | 0.99 | 1 | n.s. |
| An intelligent and responsive fresh air supply is . . . | 4.00 | 1.01 | 2 | 4.02 | 1.00 | 2 | 3.97 | 1.03 | 3 | n.s. |
| An intelligent transport system (i.e., lifts and escalators) is . . . | 3.97 | 1.04 | 3 | 3.87 | 1.09 | 3 | 4.09 | 0.96 | 2 | <0.05 |
| An intelligent and responsive thermal control is . . . | 3.88 | 0.98 | 4 | 3.84 | 1.00 | 4 | 3.92 | 0.97 | 5 | n.s. |
| An intelligent and responsive lighting is . . . | 3.86 | 0.97 | 5 | 3.84 | 1.00 | 5 | 3.89 | 0.95 | 6 | n.s. |
| Having minimum impacts on the environment is . . . | 3.83 | 0.98 | 6 | 3.74 | 0.99 | 7 | 3.94 | 0.96 | 4 | <0.05 |
| An expandable network infrastructure is . . . | 3.73 | 1.01 | 7 | 3.78 | 0.98 | 6 | 3.68 | 1.05 | 13 | n.s. |
| A real indoor green space with a variety of plants is . . . | 3.71 | 1.03 | 8 | 3.72 | 0.98 | 8 | 3.70 | 1.09 | 12 | n.s. |
| A responsive facade that can harvest daylight is . . . | 3.71 | 1.01 | 9 | 3.61 | 1.01 | 12 | 3.83 | 1.00 | 7 | <0.05 |
| An indoor social space is . . . | 3.70 | 1.04 | 10 | 3.71 | 1.02 | 9 | 3.68 | 1.07 | 14 | n.s. |
| A responsive system that can harvest solar and wind energy is . . . | 3.67 | 1.03 | 11 | 3.58 | 1.06 | 14 | 3.79 | 0.99 | 8 | <0.05 |
| A responsive acoustic environment is . . . | 3.67 | 0.89 | 12 | 3.67 | 0.90 | 10 | 3.66 | 0.89 | 15 | n.s. |
| Different social venues that facilitate interaction between users are . . . | 3.66 | 0.94 | 13 | 3.60 | 0.95 | 13 | 3.73 | 0.92 | 10 | n.s. |
| A building information system is . . . | 3.66 | 0.97 | 14 | 3.61 | 0.98 | 11 | 3.72 | 0.97 | 11 | n.s. |
| An intelligent energy system that can respond to smart grid is . . . | 3.62 | 0.98 | 15 | 3.52 | 1.00 | 15 | 3.73 | 0.95 | 9 | <0.05 |
| A responsive system that can harvest rainwater is . . . | 3.44 | 1.01 | 16 | 3.35 | 1.04 | 16 | 3.56 | 0.97 | 17 | <0.05 |
| An intelligent system that monitors people movement is . . . | 3.37 | 1.07 | 17 | 3.22 | 1.08 | 17 | 3.56 | 1.03 | 16 | <0.001 |

Note: All items were rated using a five-point Likert scale with 1 being "very unimportant" and 5 being "very important".

**Table 4.** The importance ranking of smart and sustainable buildings' features by the respondents' industry.

| Items | Commerce & Trading n = 65 | | Banking & Finance n = 56 | | Information and Comm. n = 98 | | Engineering & Construct. n = 157 | | Others n = 118 | |
|---|---|---|---|---|---|---|---|---|---|---|
| | Mean | Rank | Mean | Rank | Mean | Rank | Mean | Rank | Mean | Rank |
| An intelligent security system is . . . | 4.00 | 1 | 4.20 | 1 | 4.05 | 1 | 4.13 | 1 | 4.36 | 1 |
| An intelligent and responsive fresh air supply is . . . | 3.91 | 2 | 3.91 | 4 | 3.73 | 3 | 4.05 | 2 | 4.24 | 2 |
| An intelligent transport system (i.e., lifts and escalators) is . . . | 3.88 | 3 | 4.02 | 2 | 3.80 | 2 | 3.99 | 5 | 4.10 | 3 |
| An intelligent and responsive thermal control is . . . | 3.78 | 4 | 3.98 | 3 | 3.52 | 11 | 4.04 | 3 | 3.95 | 5 |
| An intelligent and responsive lighting is . . . | 3.72 | 6 | 3.71 | 11 | 3.73 | 4 | 4.01 | 4 | 3.92 | 6 |
| Having minimum impacts on the environment is . . . | 3.72 | 5 | 3.88 | 5 | 3.63 | 8 | 3.91 | 6 | 3.91 | 7 |
| An expandable network infrastructure is . . . | 3.65 | 9 | 3.79 | 6 | 3.64 | 7 | 3.80 | 8 | 3.75 | 12 |
| A real indoor green space with a variety of plants is . . . | 3.60 | 10 | 3.71 | 9 | 3.66 | 6 | 3.60 | 15 | 3.97 | 4 |
| A responsive facade that can harvest daylight is . . . | 3.68 | 7 | 3.73 | 8 | 3.53 | 10 | 3.71 | 12 | 3.86 | 9 |
| An indoor social space is . . . | 3.58 | 11 | 3.77 | 7 | 3.50 | 12 | 3.80 | 7 | 3.75 | 13 |
| A responsive system that can harvest solar and wind energy is . . . | 3.38 | 15 | 3.59 | 15 | 3.56 | 9 | 3.72 | 11 | 3.91 | 8 |
| A responsive acoustic environment is . . . | 3.65 | 8 | 3.71 | 10 | 3.46 | 14 | 3.66 | 13 | 3.84 | 10 |
| Different social venues that facilitate interaction between users are . . . | 3.48 | 14 | 3.64 | 13 | 3.49 | 13 | 3.73 | 10 | 3.81 | 11 |
| A building information system is . . . | 3.55 | 12 | 3.63 | 14 | 3.70 | 5 | 3.64 | 14 | 3.72 | 14 |
| An intelligent energy system that can respond to smart grid is . . . | 3.49 | 13 | 3.66 | 12 | 3.35 | 15 | 3.76 | 9 | 3.69 | 15 |
| A responsive system that can harvest rainwater is . . . | 3.38 | 16 | 3.39 | 17 | 3.15 | 17 | 3.57 | 16 | 3.58 | 16 |
| An intelligent system that monitors people movement is . . . | 3.29 | 17 | 3.45 | 16 | 3.24 | 16 | 3.38 | 17 | 3.48 | 17 |

Note: All items were rated using a five-point Likert scale with 1 being "very unimportant" and 5 being "very important".

*4.2. Factor Structure of Smart and Sustainable Buildings from the Users' Perspective*

The ANOVA results showed that three groups of respondents were identified. The first group consisted of respondents working in the commerce and trading, and banking and finance industries. An EFA was conducted with IBM SPSS Statistics 24.0 using principal component extraction with varimax rotation. The EFA results showed that the Bartlett's test of sphericity was significant at 136 degrees of freedom ($\chi^2$ = 1027.5 and significant at $p < 0.001$) and the Kaiser-Meyer-Olkin measure of sampling adequacy was 0.892, indicating that the items were suitable for factor analysis. The eigenvalue-greater-than-one rule was employed to determine the number of factors retained. Items with communalities less than 0.5, factor loadings less than 0.5, and cross-loadings greater than 0.5 were removed iteratively. The deleted items included "a responsive acoustic environment is . . . ", "an expandable network infrastructure is . . . ", "a building information system is . . . ", "an intelligent system that monitors people movement is . . . ", and "an intelligent security system is . . . " After that, 12 items were retained and three factors emerged. The three factors accounted for 68.3% of the total variance. The Cronbach's alpha values of "Smart Building Indoor Environment" (SBIE; four items), "Smart Building Skin" (SBS; four items), and "Eco and Social Spaces" (ESS; four items) were 0.85, 0.82, and 0.82, respectively. All Cronbach's alpha values were higher than the threshold level of 0.7 (Hair et al., 2006), suggesting the factors to have good reliability. Table 5 presents the factor loadings of items, eigenvalues, and percentage of variance explained by the three factors.

**Table 5.** Factor loadings of items, eigenvalues, and percentage of variance explained from EFA using responses from employees working in the commerce and trading, and banking and finance industries ($n$ = 121).

| Items | Factor | | |
|---|---|---|---|
| | **1** | **2** | **3** |
| An intelligent and responsive thermal control is . . . | 0.848 | | |
| An intelligent and responsive lighting is . . . | 0.784 | | |
| An intelligent and responsive fresh air quality is . . . | 0.729 | | |
| An intelligent transport system such as lifts and escalators is . . . | 0.687 | | |
| A responsive system that can harvest solar and wind energy is . . . | | 0.807 | |
| A responsive facade that can harvest daylight is . . . | | 0.767 | |
| A responsive system that can harvest rainwater is . . . | | 0.730 | |
| An intelligent energy system that can respond to smart grid is . . . | | 0.619 | |
| Having minimum impacts on the environment is . . . | | | 0.801 |
| An indoor social space is . . . | | | 0.785 |
| A real indoor green space with a variety of plants is . . . | | | 0.680 |
| Different social venues that facilitate user interaction are . . . | | | 0.578 |
| Eigenvalue | 3.01 | 2.62 | 2.56 |
| Variance explained (in percent) | 25.07 | 21.80 | 21.36 |
| Total variance explained (in percent) | 25.07 | 46.87 | 68.23 |

Notes: Factor 1 is named as 'Smart Building Indoor Environment' (SBIE); Factor 2 is named as 'Smart Building Skin' (SBS); and Factor 3 is named as 'Eco and Social Spaces' (ESS).

An EFA was conducted on responses from employees working in the information and communications industry. Six items including "a responsive acoustic environment is . . . " were deleted due to low communalities and high cross loadings. Table 6 shows that three factors emerged. The factors included "Eco Features and Social Spaces" (EFSS; 5 items), "Smart Building Indoor Environment" (SBIE; 3 items), and "Intelligent Information Systems" (IIS; 3 items). The Cronbach's alpha values of EFSS, IIE, and IIS were 0.76, 0.82, and 0.71 respectively.

**Table 6.** Factor loadings of items, eigenvalues, and percentage of variance explained from EFA using responses from employees working in the information and communications industry (*n* = 98).

| Items | Factor | | |
|---|---|---|---|
| | 1 | 2 | 3 |
| Different social venues that facilitate user interaction are . . . | 0.707 | | |
| A responsive facade that can harvest daylight is . . . | 0.681 | | |
| A real indoor green space with a variety of plants is . . . | 0.677 | | |
| A responsive system that can harvest rainwater is . . . | 0.668 | | |
| An indoor social space is . . . | 0.641 | | |
| An intelligent and responsive thermal control is . . . | | 0.835 | |
| An intelligent and responsive fresh air quality is . . . | | 0.785 | |
| An intelligent and responsive lighting is . . . | | 0.772 | |
| An intelligent security system is . . . | | | 0.774 |
| A building information system is . . . | | | 0.736 |
| An intelligent system that monitors people movement is . . . | | | 0.662 |
| Eigenvalue | 2.59 | 2.49 | 1.86 |
| Variance explained (in percent) | 23.55 | 22.65 | 16.88 |
| Total variance explained (in percent) | 23.55 | 46.20 | 63.08 |

Notes: Factor 1 is named as 'Eco Features and Social Spaces' (EFSS); Factor 2 is named as 'Smart Building Indoor Environment' (SBIE); and Factor 3 is named as 'Intelligent Information Systems' (IIS).

A third EFA was conducted on responses from employees working in the engineering and construction services industry and the "others". Five items including "a responsive acoustic environment is . . . " were deleted due to low communalities and high cross loadings. Table 7 shows that three factors emerged. The factors included "Smart Building Skin" (SBS; 5 items), "Eco and Social Spaces" (ESS, 4 items), and "Smart Building Indoor Environment" (SBIE; 3 items). The Cronbach's alpha values of RAE, ESS, and IIE were 0.76, 0.70, and 0.71 respectively.

**Table 7.** Factor loadings of items, eigenvalues, and percentage of variance explained from EFA using responses from employees working in the engineering and construction services, and "other" industries (*n* = 275).

| Items | Factor | | |
|---|---|---|---|
| | 1 | 2 | 3 |
| A responsive system that can harvest solar and wind energy is . . . | 0.718 | | |
| An intelligent energy system that can respond to smart grid is . . . | 0.703 | | |
| A responsive facade that can harvest daylight is . . . | 0.697 | | |
| An intelligent system that monitors people movement is . . . | 0.610 | | |
| A responsive system that can harvest rainwater is . . . | 0.608 | | |
| Different social venues that facilitate user interaction are . . . | | 0.745 | |
| An indoor social space is . . . | | 0.681 | |
| Having minimum impacts on the environment is . . . | | 0.613 | |
| A real indoor green space with a variety of plants is . . . | | 0.607 | |
| An intelligent and responsive thermal control is . . . | | | 0.819 |
| An intelligent and responsive lighting is . . . | | | 0.748 |
| An intelligent and responsive fresh air quality is . . . | | | 0.703 |
| Eigenvalue | 2.55 | 2.11 | 2.00 |
| Variance explained (in percent) | 21.28 | 17.57 | 16.62 |
| Total variance explained (in percent) | 21.28 | 38.85 | 55.47 |

Notes: Factor 1 is named as 'Smart Building Skin' (SBS); and Factor 2 is named as 'Eco and Social Spaces' (ESS); and Factor 3 is named as 'Smart Building Indoor Environment' (SBIE).

## 5. Discussion

The analyzed results highlighted the privacy paradox of smart and sustainable buildings from the users' perspective because the importance ratings showed that respondents saw an intelligent security system as the most important feature (mean = 4.16) but indicated an intelligent system that monitors people movement to be the least important feature (mean = 3.37) among the 17 items as shown in Table 3. Smart and sustainable buildings by design are fitted with smart sensors, web-enabled appliances, and surveillance systems. This type of connection creates scenarios that could be exploited by cyber criminals and create privacy-related issues, which have become a major concern in recent years. However, when the responses from males and females were examined, the mean importance rating for the item "an intelligent system that monitors people movement is . . . " of the female group (mean = 3.56) was found to be statistically significantly higher than that of the male group (mean = 3.22) at a level <0.001. Thus, females pay more attention to safety and security issues more than their male counterparts.

Although the ranking of perceived importance indicated that acoustic, indoor green, and social environment were considered as secondary features and the item "a responsive acoustic environment" was dropped in all factor analyses due to its low communality, the EFA results indicated that indoor ecology and social spaces have an impact on users' perceived smartness of smart and sustainable buildings. Building designers and owners shall focus on creating innovative, authentic ecological spaces, rather than creating non-real ecological spaces based on plastic, silk and preserved trees, flowers, and grasses. Additionally, social spaces are important to building users because people are not robots; a smart and sustainable building environment must be able to address the social needs of its users.

The results of exploratory factor analyses revealed that different user groups would focus on different factors of smart and sustainable buildings. Building users who work in the commerce and trading, banking and finance, engineering and construction, and other industries focus more on three core factors, i.e., categories of features. The factors include 'Smart Building Indoor Environment' encompassing intelligent and responsive fresh air supply, thermal control, lighting, etc., 'Eco and Social Spaces' encompassing indoor green and social spaces, and social venues, and 'Smart Building Skin' covering responsive facade or a system that harvests daylight, solar and wind energy, rainwater, and responds to smart grid design. In a highly-urbanized city such as Hong Kong, these groups of building users normally spend most of their time in their offices and meeting rooms, handling a vast amount of business transactions under pressure. Thus, a smart indoor work environment enhances the users' productivity and well-being while building's green and social spaces can improve users' work-life balance. In addition, these groups of building users also value the usefulness of 'Smart Building Skin' i.e., responsive to ambient environment in enhancing the city's sustainability. On the other hand, building users who work in the information and communication industry also perceive 'Smart Building Indoor Environment' and 'Eco Features and Social Spaces' to be important factors but consider the factor, namely 'Intelligent Information Systems', to be another core factor of smart and sustainable buildings. This result is not surprising because their profession influences what they value.

*Limitations and Future Research Direction*

The study is subject to some limitations. First, we invited part-time Master's students who had full-time jobs and asked them to invite their friends, colleagues, and relatives to participate in the study. This referral sampling technique might influence the representativeness of our samples. Nevertheless, judging from the demographic profile of respondents, we found that most respondents were aged between 20 and 29 years, had a bachelor and/or master's degree, worked in different services industries, and had more than 2 years of work experience. This group of respondents should have more interest in smart and sustainable systems, buildings, and cities. Second, the sample size was modest and there were less than 200 respondents per users' group (see Table 4). Hence, the replication of the study on a larger sample is desirable. Third, the study was conducted in

Hong Kong. As the climatic, social, and cultural conditions vary from place to place, the importance ratings of smart and sustainable building features should be generalized to other cities and regions with great caution. Finally, the study was a cross-sectional study. Further research should consider examining changes in smart and sustainable building features as technologies evolve very rapidly. Although the item "a responsive acoustic environment" was dropped due to its low communality in factor analyses, future research may explore different aspects of acoustic elements such as soundscape, noise insulation, sound absorption, and speech intelligibility in buildings.

## 6. Conclusions

Building users have different importance ratings of smart and sustainable building features. In general, they indicated that the most important features of smart and sustainable buildings were intelligent security systems and facilities management such as fresh air supply, and lifts and escalators. Acoustic, indoor green, and social environment were considered as secondary features. The EFA results showed that three core factors of perceived smartness of smart and sustainable buildings including 'Smart Building Skin', 'Smart Building Indoor Environment', and 'Eco and Social Spaces' were identified based on responses from employees working in the commerce and trading, banking and finance, engineering and construction services, and "others" industries. However, the EFA results from employees working in the information and communication industry showed that this group of users emphasized more on 'Eco Features and Social Spaces', 'Smart Building Indoor Environment', and 'Intelligent Information Systems'. Thus, the perceptions of building smartness and sustainability could be different depending on the occupation or industry of building user groups.

The human element is the key factor behind the drive to innovate and improve the effectiveness of smart and sustainable buildings. Smartness refers to making buildings better for their occupants in terms of comfort, safety, security, green environment, and community building. Technologies for such endeavors must be user-friendly and must improve the existing elements. Even the best-designed smart buildings and structures, if not properly utilized by the human component, would become unsuccessful in building smart and sustainable communities or fulfilling the goals of smart and sustainable designs. The findings of the study showed that smart and sustainable buildings should not only be responsive to user demands, they also need to address different needs from different groups of users.

**Author Contributions:** conceptualization, W.M.T. and K.H.L.; methodology, W.M.T. and L.S.L.L.; formal analysis, W.M.T.; writing—original draft preparation, W.M.T.; writing—review and editing, L.S.L.L., K.H.L., and A.W.L.C.

**Funding:** This research received no external funding.

**Conflicts of Interest:** The authors declare no conflict of interest.

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
