# Peer review of "Perceived Importance of Smart and Sustainable Building Features from the Users’ Perspective"

_smartcities, doi:10.3390/smartcities1010010_

Reviewer 1 Report

The Conclusions should contain cleraly the answers to the 3 questions fixed in the Introduction. The interviewed group is not completely representative of the building users. I suggest to enlarge the group, otherwise the information about the composition of the interviewed group should compare also in the Introduction and in the Method section. It would be better that the complete questionnaire is reported in the paper, for example in the Appendix.

English can be improved.

Author Response

Reviewer 1

1st Comment: The Conclusions should contain clearly the answers to the 3 questions fixed in the Introduction. The interviewed group is not completely representative of the building users. I suggest to enlarge the group, otherwise the information about the composition of the interviewed group should compare also in the Introduction and in the Method section. It would be better that the complete questionnaire is reported in the paper, for example in the Appendix.

Our response: Thanks very much for your comment and suggestion. In conclusion, we rewrote the first paragraph (to clearly answer the three research questions) as:

“              Building users have different importance ratings of smart and sustainable building features. In general, they indicated that the most important features of smart and sustainable buildings were intelligent security system and facilities management such as fresh air supply, and lifts and escalators. Acoustic, indoor green, and social environment were considered as secondary features. The EFA results showed that three core factors of perceived smartness of smart and sustainable buildings including ‘Smart Building Skin’, ‘Smart Building Indoor Environment’, and ‘Eco and Social Spaces’ were identified based on responses from employees working in the commerce and trading, banking and finance, engineering and construction services, and “others” industries. However, the EFA results from employees working in the information and communication industry showed that this group of users emphasized more on ‘Eco Features and Social Spaces’, ‘Smart Building Indoor Environment’, and ‘Intelligent Information Systems’. Thus, the perceptions of building smartness and sustainability could be different depending on the occupation or industry of building user groups.

Your suggestion on enlarging the sample size is very valuable. Hence, we added the following sentence to Section 5.1 as:

“…Second, the sample size was modest and there were less than 200 respondents per users’ group (see Table 4). Hence, the replication of the study on a larger sample is desirable. Third …”

Regarding the questionnaire we used, we explained more clearly in Section 3.2 with a new Table 1 in which the introduction to the study, the core dimensions, the items, and the rating scale we used are presented. Readers can have a clear view on the questionnaire we used.

2nd Comment: English can be improved.

Our response: Thanks very much for your comment. My co-authors and I reviewed the manuscript carefully and we corrected some grammatical and typing errors.

Dear Reviewer 1: Thank you so much for your valuable comments that improve our manuscript continually.

Reviewer 2 Report

-This paper deals with public perception and priority associated building services of smart buildings in the context of Hong Kong. Understanding the patterns and actual responses from the public on any technologically enhanced engineering object would always be critical. In this sense, this paper could draw attention from the potential audiences. Nevertheless, the followings need to be further elaborated.

1) More detailed and systematic explanation on how 17 smart building features were selected as representative aspects is needed. Only literature review would not make sufficient rationale for this selection.

2) Table 2 describes overall ratings with statistical values of 17 features. There are three "Overall" identical columns without any differentiating remark. It seems to be that 2 of those are for male and female ratings and the remaining 1 would be total rating but that has to be clarified by the authors.

3) Potential audiences might be interested in the interpretation on why different working sector employees responded differently when they rated 17 smart building features. The paper only explained detected resulting preference differences, and yet authors could elaborate on potential underlying reasons why those differences were revealed in such fashions.

Author Response

Reviewer 2

General comment: This paper deals with public perception and priority associated building services of smart buildings in the context of Hong Kong. Understanding the patterns and actual responses from the public on any technologically enhanced engineering object would always be critical. In this sense, this paper could draw attention from the potential audiences. Nevertheless, the followings need to be further elaborated.

Our response: Thanks very much for your comment. We would try our best to address your concerns in the revised manuscript.

 1st Specific comment: More detailed and systematic explanation on how 17 smart building features were selected as representative aspects is needed. Only literature review would not make sufficient rationale for this selection.

Our response: Thanks very much for your comment and suggestion. You are right. We should better explain why 17 smart and sustainable building features were selected. In fact, the literature review demonstrated that building features could be broadly categorized into four basic categories or dimensions. They are “building indoor environment”, “smart building skin or responsive to ambient environment”, “eco and social spaces i.e. communal factor”, and “building security and network systems”. In the revised manuscript, we added the following sentence:

“More specifically, Table 1 shows the first part of the questionnaire including an introduction to the study, the core dimensions, the items used, and the rating scale.”,

and Table 1 to Section 3.2 Design of the Questionnaire. We trust that readers can now follow why 17 smart and sustainable building features were selected.

 2nd Specific comment: Table 2 describes overall ratings with statistical values of 17 features. There are three "Overall" identical columns without any differentiating remark. It seems to be that 2 of those are for male and female ratings and the remaining 1 would be total rating but that has to be clarified by the authors.

Our response: Thanks very much for your comment. Yes, you are right. The first column should be “Overall” with a sample size of 494 while the second and third column should be “Male” (with a sample size of 272) and “Female” (with a sample size of 222), respectively. We corrected these typing mistakes in the revised manuscript.

3rd Specific comment: Potential audiences might be interested in the interpretation on why different working sector employees responded differently when they rated 17 smart building features. The paper only explained detected resulting preference differences, and yet authors could elaborate on potential underlying reasons why those differences were revealed in such fashions.

Our response: Thanks very much for your comment and suggestion. In the revised manuscript, we added the following sentences and paragraph to explain the differences we observed from the analyzed results.

“…However, when the responses from males and females were examined, the mean importance rating for the item “an intelligent system that monitors people movement is…” of the female group (mean = 3.56) was found to be statistically significantly higher than that of the male group (mean = 3.22) at a level<0.001. Thus, females pay more attention to safety and security issues more than their male counterparts.”

And

“…Building users who work in the commerce and trading, banking and finance, engineering and construction, and other industries focus more on three core factors, i.e. categories of features. The factors include ‘Smart Building Indoor Environment’ encompassing intelligent and responsive fresh air supply, thermal control, lighting, etc., ‘Eco and Social Spaces’ encompassing indoor green and social spaces, and social venues, and ‘Smart Building Skin’ covering responsive facade or system that harvest daylight, solar and wind energy, rainwater, and respond to smart grid design. In a highly-urbanized city such as Hong Kong, these groups of building users normally spend most of their time in their offices and meeting rooms, handling a vast amount of business transactions under pressure. Thus, a smart indoor work environment enhances the users’ productivity and well-being while building’s green and social spaces can improve users’ work-life balance. In addition, these groups of building users also value the usefulness of ‘Smart Building Skin’ i.e. responsive to ambient environment in enhancing the city’s sustainability. On the other hand, building users who work in the information and communication industry also perceive ‘Intelligent Indoor Environment’ and ‘Eco and Social Spaces’ to be important factors but consider the factor, namely ‘Intelligent Information Systems’, to be another core factor of smart and sustainable buildings. This result is not surprising because their professional influences what they value.”

Dear Reviewer 2: We sincere thank you for your valuable comments that improve our manuscript substantially.

Round  2

Reviewer 1 Report

Thanks for improving the paper